# A Comparison between the Egg Yolk Flavor of Indigenous 2 Breeds and Commercial Laying Hens Based on Sensory Evaluation, Artificial Sensors, and GC-MS

**DOI:** 10.3390/foods11244027

**Published:** 2022-12-13

**Authors:** Li-Bing Gao, Uchechukwu Edna Obianwuna, Hai-Jun Zhang, Kai Qiu, Shu-Geng Wu, Guang-Hai Qi, Jing Wang

**Affiliations:** Laboratory of Quality & Safety Risk Assessment for Animal Products on Feed Hazards (Beijing) of the Ministry of Agriculture & Rural Affairs, Institute of Feed Research, Chinese Academy of Agriculture Sciences, Beijing 100081, China

**Keywords:** egg flavor, sensory evaluation, electronic nose, electronic tongue, laying hens

## Abstract

The focus of this study was to compare the yolk flavor of eggs from laying hens of Chinese indigenous and commercial, based on detection of volatile compounds, fatty acids, and texture characteristics determination, using sensory evaluation, artificial sensors (electronic nose (E-nose), electronic tongue (E-tongue)), and gas chromatography-mass spectrometry (GC-MS). A total of 405 laying hens (Hy-Line Brown (*n* = 135), Xueyu White (*n* = 135), and Xinyang Blue (*n* = 135)) were used for the study, and 540 eggs (180 per breed) were collected within 48 h of being laid and used for sensory evaluation and the instrument detection of yolk flavor. Our research findings demonstrated significant breed differences for sensory attributes of egg yolk, based on sensory evaluation and instrument detection. The milky flavor, moisture, and compactness scores (*p* < 0.05) of egg yolk from Xueyu White and Xinyang Blue were significantly higher than that of Hy-Line Brown. The aroma preference scores of Xinyang Blue (*p* < 0.05) were significantly higher, compared to Hy-Line Brown and Xueyu White. The sensor responses of WIW and W2W from E-nose and STS from E-tongue analysis were significantly higher foe egg yolks of Hy-Line Brown (*p* < 0.05), compared to that of Xueyu White and Xinyang Blue. Additionally, the sensor responses of umami from E-tongue analysis, was significantly higher for egg yolks of Xueyu White (*p* < 0.05), compared to that of Hy-Line Brown and Xinyang Blue. Besides, the contents of alcohol and fatty acids, such as palmitic acid, oleic acid, and arachidonic acid, in egg yolk were positively correlated with egg flavor. The texture analyzer showed that springiness, gumminess, and hardness of Hy-Line Brown and Xueyu White (*p* < 0.05) were significantly higher, compared to Xinyang Blue. The above findings demonstrate that the egg yolk from Chinese indigenous strain had better milky flavor, moisture, and compactness, as well as better texture. The egg yolk flavors were mainly due to presence of alcohol and fatty acids, such as palmitic acid, oleic acid, and arachidonic acid, which would provide research direction on improvement in egg yolk flavor by nutrition. The current findings validate the strong correlation between the results of egg yolk flavor and texture, based on sensory evaluation, artificial sensors, and GC-MS. All these indicators would be beneficial for increased preference for egg yolk flavor by consumers and utilization by food processing industry, as well as a basis for the discrimination of eggs from different breeds of laying hens.

## 1. Introduction

Eggs are consumed globally and often considered a complete food, due to their nutrients, proteins, minerals, vitamins, and phospholipids, as well as their biological activities, which are beneficial to human health [1]. Consumers are not only interested in egg quantity, but the quality of the product, which takes priority, and eggs are the most affordable source of animal protein, compared to the other sources; thus, the need to meet the task of producing high-quality eggs. With the ever-increasing focus of consumers on the intrinsic characteristics of animal products, egg yolk flavor has received more attention. Egg flavors are reportedly bland, and the egg yolk components are rich in lipids, which contribute more to the egg flavor [2]. Researchers have been interested in understanding the intricacies of egg yolk flavor, in order to command consumer acceptance and product safety; thus, determining the organoleptic properties of egg yolk becomes crucial. It has been reported that genetic and environmental cues influence egg quality traits [3]. The literature demonstrated that the amino acid content in egg white and egg yolk were influenced by breed and genetics [4,5]. In addition, local chickens are mostly associated with genetic signatures that aid in adaptation to extreme environmental conditions that may impair animal product quality [6]. Locally adapted chickens are often associated with meat and eggs that are of higher nutritional value, compared to commercial breeds [7], and consumers have preference for eggs from indigenous chickens [8]. Nevertheless, commercial laying hens have the advantages of low genetic variability and stable egg quality [9]. The conservation of local chicken breeds for the sustenance of animal production and genetic resources biodiversity, while selecting breeds that would facilitate the production of high-quality eggs for consumers became expedient.

In China, most consumers have preference for the egg yolk flavor of indigenous laying hens, but the rationale behind the preference remains unclear. The Xueyu White and Xinyang Blue are Chinese indigenous breeds of laying hens associated with high performance [10,11]. However, there is a dearth of information on the physical and organoleptic properties of egg yolk from these breeds of laying hens. Thus, the need to examine egg yolk’s intrinsic characteristics becomes crucial, as this would act as a guide for poultry breeders and nutritionist to improve the egg yolk for consumer acceptance. Since there are different chicken breeds, the variations in their genetic make-up may bring about differences in the intrinsic traits of the eggs produced. Limited data are available on the comparison of the yolk flavor of commercial eggs from and the indigenous eggs of a Chinese breed of laying hens. We, therefore, predicted that such genetic differences may influence the egg yolk flavor of indigenous and commercial laying hens, as well as consumer preferences.

Preferences and acceptability for food are reliant, to an extent, on food taste [12], which suggests the critical role of food taste to the food industry. An evaluation of food taste could be conducted with human panels and mechanical sensors. Over time, the conventional sensory evaluation, whereby trained panelists are the machinery for evaluating food product quality using any of the five sense organs, have been adopted [13]. Sensory evaluation has been used to evaluate egg quality traits [14,15]. Sensory evaluation detected no variations on intrinsic quality of eggs from different sources, and the adoption of this approach is not without limitations [16,17]. A number of advances have been made in the fields of instrument precision, texture technology, biochemistry, and artificial intelligence over the past few years. Some systems, including electronic nose (E-nose) and electronic tongue (E-tongue), that possess the capacity to simulate human senses were developed to overcome the shortcomings of the conventional method, and the adoption of artificial sensors aids in obtaining more accurate results [18]. The E-nose consists of sampling system, sensor, and result processing and analysis, while the operating procedure is based on the interaction between semi-selective sensors and volatile substances [19]. It is designed to imitate the human nose, work with sensors that interact with volatile compounds, and process signals transmitted to the computer, and the output data are often analyzed with multivariate statistics [20]. Similarly, the E-tongue consists of sensors and electrodes that are made of lipid membranes and are attached to polymers [21]. The E-tongue senses flavors such as sour, sweet, bitter, salty, and umami, also known as the “taste sensor” [22]. There is evidence in the literature that “artificial sensors”, such as E-nose, E-tongue, and other instruments, that provide signals related to sensory evaluation attributes were used for the assessment of egg quality and internal traits. Studies have shown that the E-nose could be used to evaluate egg quality traits from the perspectives of volatile compounds [23] and egg freshness [24]. Our previous research reported that mechanical sensors could be used to evaluate the texture and sensory attributes of the albumen and yolk, fatty acid profile, and flavor of egg yolk in different breeds of chicken [25,26]. Aguinaga et al. [27] reported that sensory evaluation, E-nose and E-tongue, could be used to evaluate the influence of feed on organoleptic properties of egg, although electronic sensors reduced the analysis time and cost, compared with sensory evaluation analysis [28]. However, the results may vary due to temperature, humidity, and elusive compounds in the sample [29]. Additionally, the response obtained by each sensor only provides relatively quantitative results, and only the “odor fingerprint” of the samples can be obtained [30]. Therefore, the determination of accurate flavor substance and detection of volatile compounds and fatty acids components in egg yolk may be an operative way to discriminate the egg yolk flavor of different eggs.

Previous research has revealed that volatile compounds, such as heptanaldehyde, 6-methyl-5-heptene-2-one, octanal, and other substances, influence the egg flavor of hatched eggs, and this provided basis for discrimination of eggs from the different breeds of laying hens. [31]. Therefore, use of gas chromatography-mass spectrometry (GC-MS) might be a promising tool for the detection of volatile compounds and fatty acid determination, which may play a role in flavor composition of the egg yolks, since it could separate individual components in the mixture and conduct quantitative analysis by internal standard method [32]. However, GC-MS technology may be time-consuming, laborious, and cumbersome, and the preprocessing design may require considerable analytical skills [33]. To this end, research has been conducted to evaluate organoleptic properties of egg using a combination of sensory evaluation, E-nose, E-tongue, and GC-MS, to ensure the validity of the results. A combination of GC-MS and E-nose could be used to discriminate the fatty acid profiles and volatile compounds from egg yolks of different avian species [34] and breeds of chickens [26]. The combination of gas-chromatography-ion mobilization spectrometry (HS-GC-IMS) have been used in combination with E-nose and E-tongue to characterize the flavor attributes of egg white protein [35]. It has been reported that the sensor response values are strongly correlated with sensory values by human panelists [36]. Egg yolk flavor determination would entail the detection of various components that contribute to the flavor and flavor type, with the aid of artificial sensors and GC-MS, which would validate the results of sensory evaluation. It was unclear if the results on egg yolk flavor based on sensory evaluation would correlate with that of instrument detection. Thus, it became imperative to compare the results of egg yolk flavor from both sensory evaluation and instrument detection.

This study, therefore, sought to investigate the egg yolk flavor of three breeds (Hy-Line Brown, Xueyu White, and Xinyang Blue) of laying hens, using instruments including texture analyzer, E-nose, E-tongue, GC-MS, and sensory evaluation. We hypothesized that there existed differences in the egg yolk texture and flavor of the different breeds of laying hens, and that the results from sensory evaluation correlate with that of instrument detection. The results obtained from the study could provide a theoretical basis and technical support for instrument detection and the identification of different breeds of eggs.

## 2. Materials and Methods

### 2.1. Chemicals and Reagents

GC-grade cyclohexanone was purchased from Aladdin Reagents (Shanghai, China). Sigma-Aldrich (St. Louis, MO, USA) supplied GC-grade n-hexane. Methyl undecanoate was provided from Macklin Biochemical (Shanghai, China). HPLC-grade methanol was supplied from Mreda Technology Inc. (Beijing, China). Chemical-grade acetyl chloride and analytical-grade potassium carbonate were purchased from Sinopharm Chemical Reagent (Shanghai, China).

### 2.2. Samples Preparation

A total of 405 laying hens, 135 birds from each breed; Hy-Line Brown (50-week-old), Xueyu White (60-week-old), and Xinyang Blue (50-week-old) were used for the study. The birds were randomly allocated to to 9 replicates, with 15 birds each (1 replicate consists of 5 cages, with 3 hens each). The Hy-Line Brown and Xueyu White laying hens were obtained from Institute of Feed Research, Chinese Academy of Agricultural Science, while the Xinyang Blue layers were obtained from Mianyang in Sichuan province of China. All the laying hens were fed a corn–soybean-based diet. A total of 540 eggs (20 eggs per replicate and 180 eggs per breed) were collected when freshly laid and stored in the refrigerator at 4 ± 1 °C, although all collected eggs were used within 48 h of lay. The collected eggs were cooked with an egg cooker (Model ZDQ-B07C3, Bear Electric Co., Ltd., Foshan, China) at 100 °C for 15 min, and then the cooked yolk was separated from the egg. Then, the sample preparation for sensory evaluation followed. The yolk was put inside a bowl, then sealed with a thin plastic film to preserve the aroma. The sealed bowl with egg yolk was kept inside an incubator at 60 °C for approximately 15 min, this was to keep the yolk warm for sensory evaluation. The yolk samples were cut transversely into two halves and served to the panelists. Preparation for the frozen-dried yolk powder entails: selection of three cooked egg yolks and freezing them at temperature of −20 °C. Then, put into a freeze dryer (FD-12, Beijing Huichengjia Scientific Instrument Factory Co., Ltd., Beijing, China) for 72 h at −20 °C. All procedural protocols followed were according to that described in previous studies [26,37]. The remaining samples were used for sensor analysis by E-nose, E-tongue, and detection of volatile flavor compounds using GC-MS.

### 2.3. Sensory Evaluation

For the sensory evaluation, Goldberg’s scale method for each flavor type and preference was adopted [37]. A total of 9 panelists (four males and five females, ages 20 to 35), who had no taste disorders and had previous experience in sensory evaluation for a minimum period of 6 months, participated in the study after signing consent forms. Before the sensory evaluation, the panelists were exposed to three training sessions. In the current study, sensory evaluation was conducted by 9 trained panelists. After discussion, three aroma characteristics, nine flavor characteristics, and six texture attributes were determined. The assessment was performed 3 different times. Each test lasted for one day, with two-day break between the tests. For each test session, the panelists each were provided with 3 egg yolks (1 per breed), and each yolk was transversely cut equally into two halves. Thus, a total of 27 eggs (9 per breed) were used for each session, and all egg yolks were digitally coded. For each egg yolk per breed, the first half was used for aroma and texture assessment, and after an interval of 15 min, the second half was used for flavor assessment. During a period of 15 min, evaluators took warm water and unsalted biscuits to purify their taste. This procedure was conducted 3 times, respectively, and the eggs used were freshly boiled on the day of sensory evaluation. Egg evaluations from each treatment were replicated 3 times on 3 separate days in 1 week. Sample preparation, sensory evaluation, and analysis of mechanical sensors were conducted at sensory evaluation laboratory of the China National Institute of Standardization. The aroma, flavor, and texture characteristics of standard products used as references during the training sessions are presented in Table 1.

### 2.4. Electronic Nose Analysis

E-nose analysis was performed using a PEN3 E-nose (model PEN3, Airsense company, Schwerin, Germany) according to the method of Zhang [38], with minor modifications. The PEN3 system contained 10 metal oxide gas sensors (Table 2), which could detect olfactory cross-sensitive substances. Six grams of cooked egg yolk sample were placed in the sample bottle. The bottle was then closed, and the headspace was at 20 °C for 30 min. The sample was detected after the sensor array was cleared by air. The detection procedures were operated at 20 °C and under the following conditions: cleaning time 180 s, determination time 150 s, carrier gas flow rate 300 mL/min, and injection flow rate 300 mL/min.

### 2.5. Electronic Tongue Analysis

ARTREE II E-tongue (model ARTREE II E-tongue, Alpha MOS Inc., Toulouse, France) equipped with five test sensors and two reference sensors was used to analyze taste attributes of egg yolk. The five test sensors were SRS, STS, UMS, SWS, and BRS (Table 2). The two reference sensors were GPS and SPS. The calibration program was used to activate the sensor to recover the activity of the sensor membrane, and the procedure entailed using 1 breaker of HCl at 0.01 M program for automatic calibration, then running calibration. The cooked egg yolk sample per breed, with an approximately weight (20.00 ± 0.50 g), was added to 80 mL of deionized water and allowed to mix with the aid of an ultrasonic wave for 10 min. Then, centrifuged at 5 000 g for 10 min at 4 °C to obtain the supernatant. At 20 °C sensors were washed twice in the cleaning solution (deionized water) for 90 s and reference solutions (0.01 M HCl solutions) for 120 s, and then dipped into the sample solution for 30 s. All protocols adopted for the electronic tongue analysis were according to the method used in the study of Zhang [38]. By using taste analysis application software, the E-tongue test results were converted into taste value. Each supernatant sample was tested 4 times, but the results of the first test were discarded, so as to reduce error due to instrument.

### 2.6. Volatile Compound Analysis

The volatile compounds from egg yolk of Hy-Line Brown, Xueyu White, and Xinyang Blue were detected by GC-MS. The detection settings were proposed and modified according to Wang et al. [34]. Five grams of cooked egg yolk sample were weighed and placed in the sample vial. A total of 30 μL internal standard solution (cyclohexanone: methanol = 1:1000) was added to the sample vial. Then, in the GC-MS port, the volatile compounds were desorbed instantly by SPME equipped with a divinylbenzene/carboxen/polymethylsiloxane 50/30 mm fiber (2 cm, DVB/CAR/PDMS, gray, Supelco, Atlanta, GA, USA) at 250 °C for 5 min and then extracted at 60 °C for 40 min. The volatile compounds for analysis were performed on a GC-MS (model 7890B-5977B, Agilent Technologies, Santa Clara, CA, USA). DB-5 column (30 m × 0.320 mm × 0.25μm, Agilent Technologies, Santa Clara, CA, USA) was used in split-less mode to separate headspace volatiles with flow rates of 2.0 mL/min of helium (99.999%). The above procedures were operated under the following conditions: 40 °C oven temperature for 3 min, 2.5 °C/min ramp to 130 °C for 3 min, and 9 °C/min ramp to 250 °C for 3 min. An electron impact mass spectrometer was used to detect mass spectra between 35 and 400 m/z using electron energy of 70 eV and 230 °C source temperature. Mass spectra from the NIST 17 library (National Institute of Standards and Technology 14.L, Gaithersburg, MD, USA) and Wiley library were compared to tentatively identify volatile compounds. A match of at least 45% was considered tentative identification, and the relative concentration (ng/g) of each volatile compound was quantified. We presented the results on volatile compounds as micrograms of volatile compound per gram of yolk (micrograms/g).

### 2.7. Fatty Acid Analysis

Five grams of freeze-dried egg yolk sample was extracted according to the method of Feng [37], and then the fatty acid was methylated. The fatty acid composition and content in egg yolk were detected by GC, and detection parameters were proposed by Dong [25]. According to the test procedure, Hp-88 (100 m × 0.25 mm × 0.20 μm, Agilent Technologies, Santa Clara, CA, USA) column was used for separation, injection temperature was 250 °C, high purity helium was used as carrier, split ratio was 10:1, and flow rate was 2000 mL/min. The running program of oven temperature was maintained at 120 °C for 1 min, increased to 175 °C for 10 min at 10 °C/min, increased to 210 °C for 6 min at 3 °C/min, and increased to 230 °C for 6 min at 2 °C/min. Solvent peak removal was measured after 3 min, and the scanning range was 50–500 *m*/*z* at 220 °C for the ion source, 280 °C for the interface, and 50–500 *m*/*z* for the solvent peak removal. By determining the retention time and peak area ratio between the sample and standard (1 mg/mL methyl undecylate–hexane mixture), we were able to determine the relative content of fatty acids.

### 2.8. Texture Profile Analysis

A texture analyzer (Model TMS-PRO, Food Technology Corporation, Sterling, VA, USA) was used to determine the texture profile of the whole egg yolk. The setting for detection parameters was proposed and modified by Taskaya [39]. The starting position was 40 mm away from the detection platform. The procedural operation consists of deformation rate of 40%, detection speed of 30 mm/min, and initial force of 0.05 N.

### 2.9. Statistical Analysis

Each sample was tested 3 times, and the average value was taken for analysis. The data were analyzed by SPSS software (IBM SPSS Statistics 20; SPSS Inc., Chicago, IL, USA), and the results were expressed as average. The differences between the chicken breeds were analyzed by analysis of variance (ANOVA) and Tukey’s HSD test. When *p* < 0.05, the difference was considered significant. The data on sensory attributes of egg yolk from three breeds of laying hens were analyzed using two-way ANOVA. The model included breed and panelists as fixed effects and two-way interactions of panelist by breed. Pearson correlation analysis and PCA were performed based on dimensionality reduction (Origin Pro, version 2021. Originlab Corporation, Northampton, MA, USA). The PLS (PLS, Xlstat version 2016, Addinsoft Inc., New York, NY, USA) regression analysis was used to study the relationship between sensory evaluation (y variable) and detection (x variable).

## 3. Results

### 3.1. Sensory Evaluation of Egg Yolk

The results on the analyzed differences on sensory evaluation among the three breeds of laying hens are presented in Table 3. All sensory attributes were significantly different across (*p* < 0.05) the breeds, although there were no significant variations for egg aroma, salty flavor, stickiness, and moisture (*p* > 0.05). The egg flavor, salty flavor, moisture, compactness, and hardness scores of the egg yolk of Xinyang Blue and Xueyu White were significantly higher (*p* < 0.05) than the scores from egg yolks of Hy-Line Brown, while the ammonia aroma was significantly lower (*p* < 0.05). Between the Xinyang Blue and Xueyu White, the egg aroma, egg flavor, compactness, and hardness of the egg yolk were significantly lower (*p* > 0.05), while the salty flavor and moisture were higher (*p* < 0.05) for Xueyu White, compared to Xinyang Blue. In addition, the aroma preference of Xinyang Blue was significantly higher (*p* < 0.05), compared to that of Hy-Line Brown and Xueyu White. Pearson correlation for the preferences for egg yolk are presented in Table 4. There was a correlation between overall preference and flavor and texture preference. The studied phenomenon was presented as follows.

### 3.2. Electronic Nose Analysis of Egg Yolk Aroma

The results obtained from E-nose analysis on the egg yolk aroma are presented in Figure 1A–C. The result analysis showed that Xueyu White and Xinyang Blue had significantly lower sensor response values in W1W and W2W, compared to that of Hy-Line Brown (*p* < 0.05; Figure 1A). In PCA analysis, the PC1 contributed 63.7% (Figure 1B), while the PC2 contributed 20.3% (Figure 1C). Analysis showed that for PC1, W1S and W2S sensors contributed more, whereas for PC2, a greater contribution was made by W6S sensors. The findings on the analysis of egg yolk aroma using E-nose sensors based on PC1 (Figure 1B) showed non-distinct separation of points among the breeds, and this may suggest a lack of capacity to discriminate the difference in egg yolk aroma among the Hy-Line Brown, Xueyu White, and Xinyang Blue. Further, as shown in Figure 1D, W5S and W1W sensors were positively correlated with egg yolk aroma, which may indicate that egg yolk aroma may be due to the content or presence of aromatic substances. Additionally, the W2S sensors had a high response intensity to fishy and ammonia aroma, and the fishy and ammonia aroma might be influenced by alcohols and aldehydes content.

### 3.3. Electronic-Tongue Analysis of Egg Yolk Flavor

The results for egg yolk flavor were shown in the radar map (Figure 2A). The results from STS and UMS sensors showed that there was significant variation (*p* < 0.05) in the egg yolk favor among the breeds of laying hens. According to PCA analysis, the two principal components account for 88.3% of the total variance, an indication that they were sufficient to represent most of the information on the flavor characteristics of the egg yolk sample, with PC1 and PC2 accounting for 59.4% and 28.9%, respectively (Figure 2B). In the score plot, there were distinct separate distances for the sensor results among the breeds, this suggests that E-tongue could discriminate the egg yolk flavor of the breeds. The SWS and BRS sensors had the largest contribution to the PC1, while STS sensors had the largest contribution to the PC2 (Figure 2C). In addition, the correlation between the sensors and characteristic flavor of the egg yolk are shown in Figure 2D; the STS sensor was negatively correlated with egg flavor, the BRS sensor had a positive correlation with ammonia flavor, and the UMS sensor had a positive correlation with sweet and salty flavor. These findings suggest that results from E-tongue sensors are strongly correlated with that of sensory evaluation of egg yolk flavor.

### 3.4. Volatile Compound Analysis of Egg Yolk

After the removal of outliers (samples with less than 45% acceptance score), we identified 56 volatile compounds in egg yolks, including alkanes, ketones, acids, esters, aldehydes, S, N-containing compounds, aromatics, and alcohols (Appendix A). Approximately total of 42, 48, and 44 volatile compounds were detected in Hy-Line Brown, Xueyu White, and Xinyang Blue of yolks, respectively. Furthermore, eight volatile compounds were significantly different among the breeds (Appendix A). The PLS loading plot (Figure 3) showed that alcohols were positively correlated with sensory evaluation scores on egg yolk aroma (*p* < 0.05). The sensory evaluation scores for fishy and ammonia aroma, respectively, were positively correlated to contents of esters and acids, but negatively correlated with content of ketones and aromatic substances (*p* < 0.05).

### 3.5. Fatty Acids of Egg Yolk

The results of fatty acid contents of egg yolks from Hy-Line, Xueyu, and Xinyang laying hens were listed in Table 5. Breed differences (*p* < 0.05) were notable for palmitoleic acid (PALMO, C16:1), oleic acid (OA, C18:1), arachidonic acid (AA, C20:4), docosapentaenoic acid (DHA, C22:6), and total n-3 polyunsaturated fatty acid (PUFA) content of the egg yolk. The fatty acids contents in egg yolks of Xinyang Blue hens had significantly higher (*p* < 0.05) contents of PALMO and AA, compared to that found in the egg yolk of Xueyu White, whereas the DHA in the egg yolk of Hy-Line Brown hens were significantly higher (*p* < 0.05) than that of egg yolk from Xinyang Blue. Additionally, the total n-3 PUFA of Hy-Line Brown and Xueyu White were significantly increased (*p* < 0.05), compared to that of Xinyang Blue. In addition, the results from PLS analysis showed that egg yolk flavor was positively correlated with PALM, OA, and AA, but negatively correlated with PALMO, ALA, and LA (Figure 4), whereas ammonia flavor was negatively correlated with n-3 PUFA and DHA. Fatty acids (oleic and arachidonic) had a notable influence on the flavor of the egg yolk. The findings on egg yolk flavor from PLS sensory evaluation analysis were consistent with results of the E-tongue sensor analysis.

### 3.6. Texture Profile Analysis of Egg Yolk

The results on texture characteristics of egg yolks from the different breeds of laying hens detected by texture analyzer are presented in Table 6. Breed differences were observed for springiness, gumminess, chewiness, and hardness of the egg yolk (*p* < 0.05), whereas no breed differences were notable for cohesiveness (*p* > 0.05). It showed that the springiness, chewiness, and hardness of egg yolk from Hy-Line Brown and Xueyu White were significantly higher (*p* < 0.05), compared to that from Xinyang Blue hens. The gumminess of egg yolk from Xinyang Blue hens was significantly lower (*p* < 0.05), compared to that of Xueyu White and Hy-Line Brown, whereas egg yolk of Xinyang Blue hens had the lowest elasticity and hardness, compared to that of other breeds (*p* < 0.05). Interestingly, in PLS analysis (Figure 5), the results demonstrated that the sensory characteristics of adhesive dentition were related to hardness and cohesiveness, but not significantly (*p* > 0.05). (Figure 5).

## 4. Discussion

The Xueyu White and Xinyang Blue are Chinese indigenous breeds and are local-ly reared, while the Hy-Line Brown is used for commercial purposes. The assessment of egg yolk flavor preference based on sensory evaluation indicated that there were significant differences in flavor preference, texture preference, and overall preference. Furthermore, previous studies mostly focused on evaluation of egg aroma, flavor, texture, and overall preference on a singly basis [27,37,40]. In the present study, all the variables were examined together to provide a basis for results validation. The overall preference for egg yolk flavor was highly positively correlated with texture, flavor, and aroma preference. This suggests that breed differences contributed to variation in the results of egg yolk flavor preference, while texture and flavor were the main factors affecting preference for egg yolk flavor.

The egg yolk compactness and moisture scores of Xinyang Blue and Xueyu White were better than that of Hy-Line Brown. The changes in the texture of egg yolk from various breeds might be a function of molecular and structural changes in the yolk components in response to heat (boiling). Heating influences the distribution of egg yolk lipids, which is often influenced by egg weight, but egg weight was not significantly different in this study. Therefore, there may be another rationale to explain the changes in egg yolk texture. Egg yolk consists mainly of lipoprotein molecules (combined with protein and lipid); therefore, lipoprotein composition and structure exert significant influence on yolk texture [41]. Additionally, the lipid oxidation of egg yolk results in the dissolution of lipids, and that enhances the springiness of egg yolk texture [42]. During the heating process, lipids migration occurred in steamed eggs to form high-fat and low-fat structures, respectively. In the high-fat structure, the protein molecules are unfolded, and the crystallization rate becomes low, thus stabilizing the egg yolk texture, while increasing the hardness [31]. It has been reported that the addition of oil to egg yolk causes a change in the egg yolk lipoprotein structure and texture [42]. Further on, moist relates to the moisture content in the egg yolk, and moisture was positively correlated with textural hardness and cohesiveness in this study. Prior to boiling of the egg yolk samples, we tested the water content of the raw egg yolk (51.73 ± 2.08), and there were no significant variations among the egg yolk samples for water content. Therefore, the positive correlation between egg yolk moisture and egg yolk cohesiveness was not due to water content. Additionally, the size of surface pores of boiled egg yolk particles images viewed under scanning electron microscopy were large, due to increased cohesiveness, as a result of water evaporation during heating [43]. These changes can also be explained by variations in protein structure: alterations in the secondary structure of proteins exposed the hydrophobic bonds, and as such, led to formation of loose structures, which increased the water holding capacity [44]. Moreover, it could be inferred that variations in compactness and moisture traits based on sensory evaluation could be attributed to the differences in textural properties, i.e., hardness, springiness, gumminess, and chewiness, as well as the composition and structure of yolk lipids.

The overall flavor characteristics may be determined, to an extent, by aroma and flavor attributes, which must be appealing to the consumers for acceptability. In this study, the egg yolk flavor of the Xinyang Blue and Xueyu White had stronger milky flavor, compared to that of Hy-Line Brown. However, the salty flavor of egg yolk from Xueyu White was more obvious, compared to that of other breeds. The variation in the milky flavor of egg yolks may be due to fatty acid content. There is evidence in the literature that the milky flavor of egg yolk may be influenced by fatty acid content in the egg yolk. Reports demonstrated the enhancement of milky flavor of egg yolk, due to the decomposition of stearic acid, oleic acid, and linolenic acid, due to heat treatment [45], increased the content of oleic acid in the egg yolk, due to dietary effect of hempseed and hempseed oil [46], and increased content of oleic acid and arachidonic acid [37]. In this study, the higher milky flavor score of egg yolk from Xinyang Blue and Xueyu White breeds may be explained by the variation in the egg yolk content of oleic acid and arachidonic acid. Salty flavor is generally the taste of a salt solution. The salty flavor score of the egg yolk was positively correlated with the response value of the E-tongue umami sensor and was negatively correlated with the n-3 PUFA content. Recent research had shown that saltiness perception was affected by the increase in umami sensors [47]. Therefore, the significant salty flavor scores of egg yolk from Xueyu White could be linked to alterations in n-3 PUFA content and sensitivity of the umami sensors. Invariably, the umami sensors may be influenced by the composition and content of amino acids and lipids, which was the characteristic flavor of egg yolk [48]. Additionally, umami sensory scores were regulated by arachidonic acid, due to its influence on cation channels [49], while the umami flavor could be affected by the composition and content of amino acids and 5’- nucleotides [38]. Furthermore, the E-tongue umami sensor mainly uses the electrostatic interaction between the lipid film and the odorant to generate potential changes and output them as visual data [19]. Instrumental detection and sensory evaluation had different perceptions of umami in the current study in same lieu, and the study of Aguinaga et al. [27] showed that sensory evaluation could not distinguish the flavor difference of eggs produced by adding yeast biomass in feed, but the instrument could distinguish the flavor and aroma of eggs. Taken together, the salty flavor score of egg yolk was positively correlated with the response value of the E-tongue umami sensor and may be related to the content of arachidonic acid. The breed differences for milky flavor are explained by the variation in the egg yolk content of fatty acids (oleic acid and arachidonic acid) among the breeds. In order to further establish the components and characteristic that contributed to the egg yolk flavor, we determined the aroma, flavor characteristics, and volatile compounds that may contribute to egg yolk flavor using sensory evaluation and instrument detection.

Compared with the Xinyang Blue, the aroma preference score of Hy-Line Brown and Xueyu White was higher, and the score of egg aroma increased, but the difference was not significant. The loading diagram showed that the score of egg aroma was positively correlated with the content of alcohol volatile substances and the response value of oxygen and nitrogen compounds sensor in the E-nose. Alcohols are one of the main components of egg yolk volatiles, with mushroom flavor, fruit flavor, and grass flavor [31,50]. One study reported that, when the threshold value of alcohol as a volatile was low, the aroma activity value was high [51]. As a characteristic aroma substance, 1-octene-3-ol could effectively be used to distinguish products from chicken breeds varieties and length of exposure to smoking [50,52]. Studies have shown that there was a good correlation between sensory evaluation and GC-MS detection data [53]. The results of this experiment showed that alcohol substances were positively correlated with egg aroma. In addition, the E-nose sensor response values of aromatic organosulfur compounds were positively correlated with egg aroma. Conversely, sulfide is one of the main components of egg yolk volatiles, and dipropyl trisulfide was detected in eggs, but it was not present in egg yolks and was not easily detected [54]. The response value of the E-nose sensor was more sensitive than the sensory evaluation, and the combination of the two provides data supported for food research and development [24]. Therefore, egg aroma scores were positively correlated with alcohol and amine content. According to PCA analysis, there was a high correlation between sensory evaluation and instrument detection, but the underlying mechanism still warrants further investigation.

## 5. Conclusions

Overall, our data showed that there were differences in the aroma, flavor, and texture of egg yolks of Hy-Line Brown, Xueyu White, and Xinyang Blue. The texture and flavor of egg yolk were the main factors affecting the preference for egg yolk flavor in different breeds. The enhanced egg yolk flavor was associated with increased content of alcohols, oleic acid, and arachidonic acid. The increased preference for egg yolk of the indigenous breeds accrued from high scores for milky flavor, moisture, texture, and compactness. Therefore, the breed difference in overall preference, yolk flavor, and texture suggested the superiority of the indigenous breeds over the exotic breed. The findings in this study provide a basis for the evaluation of yolk flavor using sensory evaluation and instrument detection and contribute to the breeding and nutrition regulation of laying hens.

## Figures and Tables

**Figure 1 foods-11-04027-f001:**
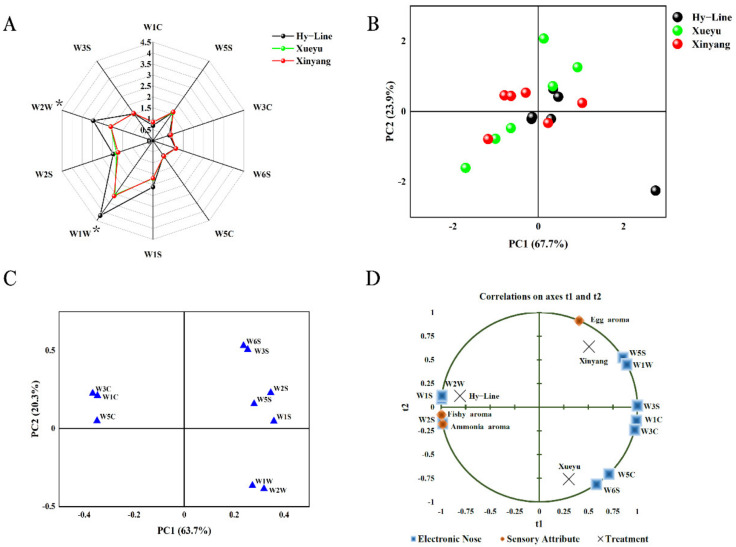
Presents the E-nose analysis on aroma of boiled egg yolk from Hy-Line, Xueyu, and Xinyang laying hens; (**A**). Radar chart of the E-nose response data, (**B**) Score plot of egg yolks (**C**). PCA loading plot of the E-nose response data, and (**D**) an overview of the correlation loadings from PLS analyses with aroma compounds detected by the E-nose as X-variables and sensory attributes as Y-variables. Note: * Means significant difference (*p* < 0.05). Hy-Line: Hy-Line Brown; Xueyu: Xueyu White; Xinyang: Xinyang Blue.

**Figure 2 foods-11-04027-f002:**
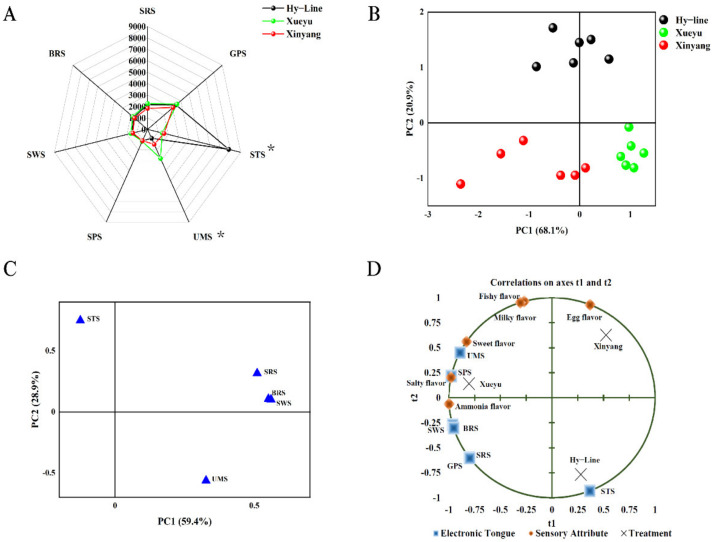
Presents the results of E-tongue sensors on flavor of boiled egg yolk from Hy-Line, Xueyu, and Xinyang laying hens; Radar chart of the E- tongue response data (**A**), score plot (**B**) of egg yolks, PCA loading plot (**C**) of the E- tongue response data and an overview of the correlation loadings (**D**) from PLS analyses with flavor compounds detected by the E- tongue as X-variables and sensory attributes as Y-variables. Note: * Means significant difference (*p* < 0.05). Hy-Line: Hy-Line brown; Xueyu: Xueyu White; Xinyang: Xinyang blue.

**Figure 3 foods-11-04027-f003:**
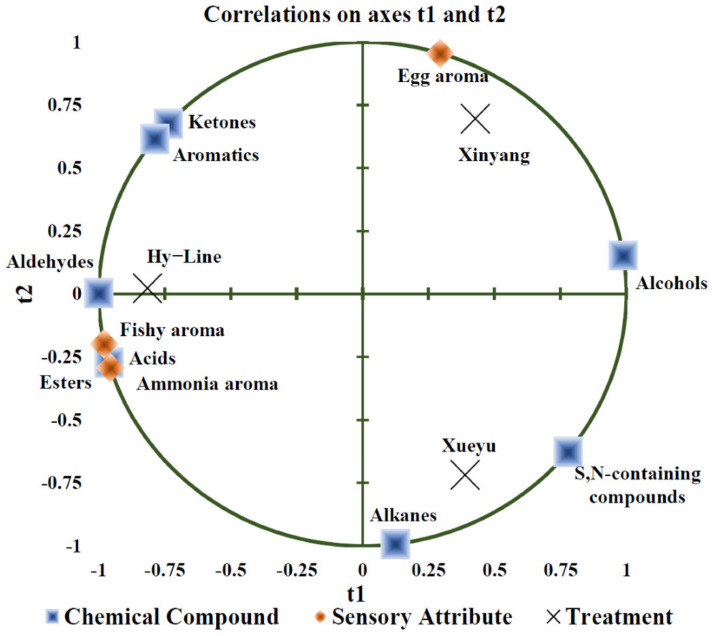
An overview of the correlation loadings from PLS analyses with volatile compounds as X-variables and sensory attributes as Y-variables. Note: Hy-Line: Hy-Line brown; Xueyu: Xueyu White; Xinyang: Xinyang blue.

**Figure 4 foods-11-04027-f004:**
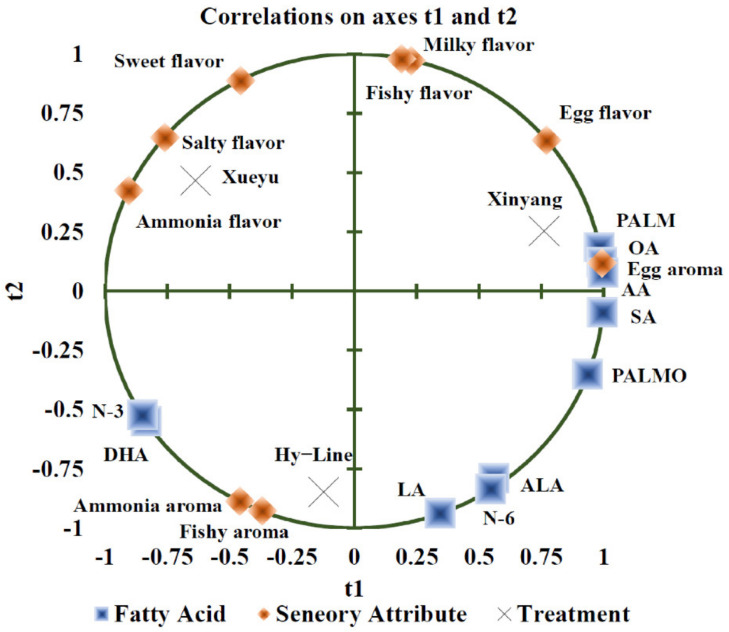
An overview of the correlation loadings from PLS analyses with fatty acid as X-variables and sensory attributes as Y-variables. Note: Hy-Line: Hy-Line Brown; Xueyu: Xueyu White; Xinyang: Xinyang Blue. PALM: Palmitic acid; PALMO: palmitoleate acid; SA: stearic acid; OA: oleic acid; LA: linoleic acid; AA: arachidonic acid; ALA: α-linolenic acid; DHA: docosapentaenoic acid; N-3: ALA + DHA; N-6: LA + AA.

**Figure 5 foods-11-04027-f005:**
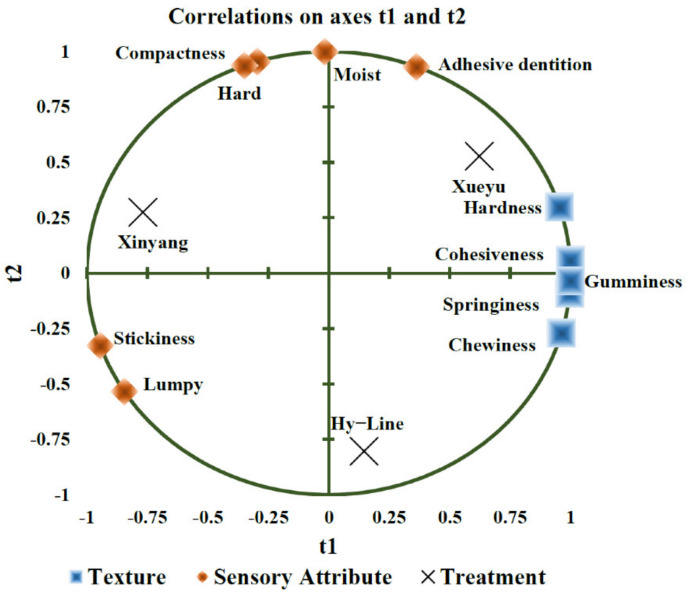
An overview of the correlation loadings from PLS analyses with texture as X-variables and sensory attributes as Y-variables. Note: Hy-Line: Hy-Line Brown; Xueyu: Xueyu White; Xinyang: Xinyang Blue.

**Table 1 foods-11-04027-t001:** Aroma, flavor, and texture standard products used as reference in training sessions.

Attribute	Standard Products/Amount
Aroma	
Egg aroma	Blended commercial egg, cooked (CP Group, Beijing, China)
Ammonia aroma	Ammonia flavor liquor nose (Le Nez du Vin, France)
Fishy aroma	Raw fresh menhaden fillets (Wumarket, Beijing, China)/15 g
Flavor	
Egg flavor	Blended commercial egg, cooked (CP Group, Beijing, China)
Fishy flavor	Raw fresh menhaden fillets (Wumarket, Beijing, China)/15 g
Ammonia flavor	Ammonia flavor liquor nose (Le Nez du Vin, France)
Milky flavor	1.26% low-fat milk (CP Group, Beijing, China)/5 g
Sweet flavor	Sucrose/1 g
Salty flavor	0.25% salt solution (Wumarket, China National Salt Industry Group, Xinjiang, China)/1 g
Texture	
Adhesive dentition	The stickiness felt by the teeth when biting the egg yolk
Stickiness	The tongue felt sticky from egg yolk
Moisture	The tongue felt moisty from egg yolk
Lumpy	The tongue felt the graininess of the egg yolk when gently sipping the egg yolk
Compactness	The hands and mouth felt compactness of yolk felt
Hard	Resistance of teeth during occlusion

**Table 2 foods-11-04027-t002:** Electronic nose sensors and electronic tongue sensors and their performance description.

Items	Sensors Name	Sensor Characteristics
E-nose		
1	W1C	Aromatic
2	W5S	Nitrogen oxides
3	W3C	Ammonia
4	W6S	Hydrogen
5	W5C	Alkane
6	W1S	Methane
7	W1W	Sulfur
8	W2S	Alcohol, aromatic
9	W2W	Aromatic, sulfur organic
10	W3S	High concentrations > 100 ppm
E-tongue		
1	SRS	Sourness
3	STS	Saltiness
4	UMS	Umami
6	SWS	Sweatiness
7	BRS	Sourness

**Table 3 foods-11-04027-t003:** Sensory attributes of egg yolks from Hy-Line, Xueyu, and Xinyang laying hens.

Attribute	Hy-Line ^1^	Xueyu ^1^	Xinyang ^1^	SEM ^2^	Source of Variation (*F*-Value)	
					Breed	Panelist
Aroma						
Egg aroma	8.73 ^b^	8.70 ^b^	10.46 ^a^	0.144	5.55 *	0.84
Ammonia aroma	1.32 ^a^	0.99 ^b^	0.57 ^b^	0.110	5.28 *	5.99 *
Fishy aroma	1.57	1.63	1.16	0.918	1.88	6.12 *
Flavor						
Egg flavor	8.94 ^c^	9.47 ^b^	10.10 ^a^	1.675	3.29 *	9.38 *
Fishy flavor	0.76	0.75	0.68	0.470	1.67	10.82 *
Ammonia flavor	0.65	0.73	0.51	0.425	0.62	3.70 *
Milky flavor	5.54	7.65	7.61	1.537	12.83	2.80 *
Sweet flavor	1.58	2.06	2.03	1.075	0.42	17.14 *
Salty flavor	1.42 ^c^	2.98 ^a^	1.94 ^b^	1.053	7.24 *	1.54
Texture						
Adhesive dentition	6.30	6.66	6.62	2.003	0.61	25.10 *
Stickiness	6.12	5.09	5.87	1.861	1.14	2.94
Moisture	3.84 ^c^	5.64 ^a^	5.08 ^b^	1.198	6.92 *	0.53
Lumpy	2.91	2.39	2.75	1.575	0.61	7.13 *
Compactness	6.85 ^c^	9.53 ^b^	9.97 ^a^	1.951	29.42 *	6.62 *
Hard	2.13 ^c^	2.83 ^b^	3.34 ^a^	1.245	3.96 *	3.96 *
Preference						
Aroma preference	9.86	9.75	10.74	2.047	1.38	4.00 *
Flavor preference	9.53 ^b^	10.07 ^b^	10.81 ^a^	1.714	4.72 *	9.64 *
Texture preference	9.07	10.13	10.24	2.284	3.62	17.19 *
Overall preference	9.36	10.30	10.79	2.225	5.22	18.07 *

Interaction F-values and significance not shown in Table 2 (Breed *n* = 3; Panelist *n* = 9); Breed x Panelist 8.84 *; Ammonia aroma 1.79 NS; Egg flavor 6.48 *; Salty flavor 11.96 *; Moisture 4.94 *; Compactness 7.83 *; Hard 6.94 *; Flavor preference 2.10 *. ^1^ Hy-Line, Hy-Line Brown. Xueyu, Xueyu White. Xinyang, Xinyang Blue, the below is the same. ^2^ SEM, standard error of the mean, the below is the same. ^a–c^ Means within a row with no common superscripts differ significantly (*p* < 0.05), and the below is the same.

**Table 4 foods-11-04027-t004:** The Pearson correlation analysis of sensory evaluation preference.

	Aroma Preference	Flavor Preference	Texture Preference	Overall Preference
Aroma preference	1.00			
Flavor preference	0.44	1.00		
Texture preference	0.45	0.76	1.00	
Overall preference	0.49 *	0.77 **	0.99 **	1.00

* Means significant difference (0.01 < *p* < 0.05); ** means extremely significant difference (*p* < 0.01).

**Table 5 foods-11-04027-t005:** Fatty acid of egg yolks from Hy-Line, Xueyu, and Xinyang laying hens (mg/g).

Items	Hy-Line ^1^	Xueyu ^1^	Xinyang ^1^	SEM ^2^	*p*-Value
PALM (C16:0)	74.78	70.56	78.93	1.97	0.27
PALMO (C16:1)	11.47 ^ab^	7.71 ^b^	13.23 ^a^	0.83	0.01
SA (C18:0)	35.34	32.31	32.97	1.21	0.57
OA (C18:1 n9)	7.22	6.28	9.97	0.71	0.08
LA (C18:2 n6)	55.08	42.6	48.6	2.67	0.16
AA (C20:4 n6)	5.43 ^ab^	4.5 ^b^	7.64 ^a^	0.52	0.03
ALA (C18:3 n3)	4.74	4.78	N.D.^3^	0.44	0.96
DHA (C22:6 n3)	5.55 ^a^	4.49 ^ab^	2.25 ^b^	0.43	0.02
Total n-3 PUFA	10.29 ^a^	8.48 ^a^	2.25 ^b^	0.99	0.01
Total n-6 PUFA	60.52	47.1	56.25	2.98	0.18

Abbreviations: PALM, Palmitic acid; PALMO, palmitoleate acid; SA, stearic acid; OA, oleic acid; LA, linoleic acid; AA, arachidonic acid; ALA, α-linolenic acid; DHA, docosapentaenoic acid; Total n-3, ALA + DHA; Total n-6, LA + AA. ^1^ Hy-Line, Hy-Line Brown. Xueyu, Xueyu White. Xinyang, Xinyang Blue, the below is the same. ^2^ SEM, standard error of the mean, the below is the same. ^3^ N.D., not detected. ^a,b^ Means within a row with no common superscripts differ significantly (*p* < 0.05), while means without alphabets indicate no significant difference.

**Table 6 foods-11-04027-t006:** Texture of egg yolks from Hy-Line, Xueyu, and Xinyang laying hens.

Items	Hy-Line ^1^	Xueyu ^1^	Xinyang ^1^	SEM ^2^	*p*-Value
Cohesiveness	0.43	0.45	0.40	0.02	0.50
Springiness (mm)	5.20 ^b^	5.70 ^a^	3.80 ^c^	0.16	<0.01
Gumminess (N)	1.83 ^ab^	2.07 ^a^	1.31 ^b^	0.12	0.03
Chewiness	7.76 ^a^	8.03 ^a^	4.53 ^b^	0.45	<0.01
Hardness (N)	3.87 ^b^	4.80 ^a^	3.27 ^c^	0.17	<0.01

^1^ Hy-Line, Hy-Line Brown. Xueyu, Xueyu White. Xinyang, Xinyang Blue, the below is the same. ^2^ SEM, standard error of the mean, the below is the same. ^a–c^ Means within a row with no common superscripts differ significantly (*p* < 0.05), and below is the same.

## Data Availability

All data included in this study are available by contacting the corresponding author.

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
