# Peer review of "A Comparison between the Egg Yolk Flavor of Indigenous 2 Breeds and Commercial Laying Hens Based on Sensory Evaluation, Artificial Sensors, and GC-MS"

_foods, 2022, doi:10.3390/foods11244027_

Round 1

Reviewer 1 Report

Dear authors,

This manuscript investigated an important comparison between the eggs produced from indigenous laying breeds and commercial laying lines based on sensory evaluation, electronic nose, electronic tongue, volatile flavor substances, fatty acids and texture characteristics of egg yolk. It is interesting research that adds to the knowledge and fits the journal’s scope. There are some minor revisions need to be adjusted before the final acceptance:

L.11: “indigenous breeds”. Please add “laying” after “indigenous”.

L.136: “Each breed has 9 replicates”. Change “breed” to “strain” or “genotype” to be more accurate.

L.424-426: I suggest adding references to support this statement.

L.426: “.. fatty acid”. Please add “S” to acid.

L.432-433: “They were one of the main components of egg yolk volatiles, with mushroom flavor, fruit flavor and grass flavor”. Would you please make it clearer?

Table 2: Add number “1” after Xueyu and Xinyang, such you made with the Hy-Line.

Please, do the same addition to Table 3 and 4.

Table 3: Please revise the significance of PALMO (C16:1).

Conclusions: L.5: “.. which was beneficial to ..”, was or were?

Author Response

Special thanks to you for your scholarly comments on our manuscript. The comments provided are valuable and very helpful for revising and improving the quality of our paper. We have studied the comments carefully and have made the necessary corrections. We have done the proof reading of the manuscript and we therefore re-submit a new version. The primary corrections in the paper and the response to the reviewer’s comments are as follows.

Reviewer 2 Report

A Comparison Between the Egg Yolk Flavor of Indigenous 2 Breeds and Commercial Laying Hens Based on Sensory Evalua-3 tion, Artificial Sensors and GC-MS

The study focuses on comparing the sensory characteristics, the instrumental texture, the volatile compounds, the fatty acids and the response measured with an electronic nose and one electronic tongue of egg yolk from two local breeds and a commercial or foreign one.

This is an interesting topic, and the comparison includes several analytical methods. However, the experimental design is not suitable to research differences among breeds. The number of parents is missing, and only nine replicates per breed were used to compare the breeds. It is not clear what was compared and what was a replicate: apparently, there were 9 hens per breed, but it is unknown what the genetic design was and if the 9 hens were representative of each breed (perhaps there were nine hens, and perhaps the nine were sisters or half-sisters). The Material and methods section is incomplete, with critical information missing, which makes difficult to understand the results. In addition, there are a considerable number of mistakes in the document.

Specific comments

Introduction

-          The introduction section is lengthy and does not mention why the yolk but not the white was studied and why those breeds were chosen (and there is not information about those breeds). Some sentences are hard to understand (for example lines 99-102), and there is confusing information, for example lines 40-41 indicates “In China, most consumers have preference for egg from indigenous local chickens due to its flavor.” but lines 53-55 “However, the use of exotic breeds as commercial laying hens tends to act as a driver for reduction in genetic variation while the egg quality of hybrids tends to be superior [8]”: are “exotic”, “hybrids”, and “commercial” hens the same?

-          Line 73: Replace “stimulate” with “simulate”

-          Line 109. Define “IMS”.

-          Lines 111-113. The sentences scould be removed since they do not seem to be related to the previous sentences, and it is not clear why they are relevant.

Materials and Methods

-          Paragraph 2.1: formulas and CAS numbers are not relevant as long as the IUPAC names are provided.

-          Lines 132-133. The mean weight of the eggs should be provided since differences in weight for eggs with the same heating treatment obviously can explain most differences in texture and flavour. This should be mentioned in the discussion.

-          Lines 132-133. The number of parents should be included.

-          Lines 135-136: “white and blue eggs respectively” should be removed since it was already indicated in lines 132-133.

-          Line 136. “Each breed has 9 replicates” should be rewritten to make clear what was a replicate: were there 9 hens, and were all the eggs from each hen (20 eggs per hen) a replicate? It should be indicated how the eggs were kept and handled: were they kept under refrigeration and analysed together (with fresh eggs and refrigerated ones together)?

-          Line 139. The cooking temperature should be included.

-          Line 140. It should be indicated what was “The 10 pieces of cooked egg yolk”. How were they “sealed”? Why were they placed in an incubator? For how long?

-          Line 142. It should be indicated what was “a fraction” of each sample (weight or volume), and how was it cut.

-          Line 143. The freeze-dryer details should be provided.

-          Figure 1. The figure should be removed since the appearance of the eggs is not relevant for the study.

-          Lines 150-151. Reference [35] does not include any information about preference and texture evaluation, and the descriptive traits are not the same as those included in the study. A relevant definition of the traits should be included. The choice of terms should be consistent. Preference should have been assessed by using consumers, not trained panellists.

-          Line 154. For clarity, rewrite “the panelists were offered three sessions of pieces of training”.

-          Line 156. Texture and preference should also be included.

-          Line 160. “digitally coded” was already included in line 155.

-          Line 161-162. There is something missing in the sentence.

-          Line 164. More details should be included: total number of eggs assessed per breed, eggs assessed per panellist, samples assessed per panellist per session, number of sessions per day, and number of days with sessions.

-          Line 172 and 175. The temperature should be included.

-          Line 192. Replace “identified” with “tentatively identified”. The retention indexes should be used for the tentative identifications.

-          Lines 196 and 198. Was it ng/g or micrograms/g?

-          Lines 205-206. Rewrite for clarity.

-          Line 209. Time and temperature should be provided.

-          Line 212. It should be clear what was measured four times: was the same supernatant prepared once and measured four times, or was the supernatant prepared and measured four times?

-          Line 216. A reference for the transesterification method should be included.

-          Line 227. The name and concentration of the standard should be included.

-          Line 235. It is indicated that “Each sample was tested 3 times and the average value was taken for analysis”, but in line 212 it is indicated that each sample was measured four times.

-          Line 238. More information on the sensory data analysis should be included (two-way ANOVA?), and the significance of the panellist effect (and if possible the session effect) should be included in the Results and Discussion sections.

-          Line 239. The name of the correlation test (Pearson, Spearman?) should be included (also in the Results and Discussion sections).

Results

-          Figure 2. The figure is difficult to read, and could be easily replaced with text. The correlation test should be indicated in the title.

-          Line 267. Replace 72.1% with 63.7%.

-          Line 268. Replace 15.7% with 20.3%.

-          Figure 3 and 4. The radar chart should include the statistical significance, and an error measurement if possible (the same applies for other radar charts).

-          Line 285. Define what an outlier (criterion) was.

-          Line 290. Replace PCA with PLS.

-          Lines 304-306. The percentages are incorrect.

-          Line 307. They were not “negatively related” since the loading were insignificant. The same occurs with other figures.

-          Line 310. It is not clear what “selected egg yolks” means.

-          Line 321. The data do not support the conclusion included in the sentence. The study and the statistical analysis focused on global relationships, but not on cause-effect relationships. Rewrite the sentence.

-          Line 345. “elasticity” is not included in any table.

-          Line 351. Replace PCA with PLS.

-          Lines 351-353. Rewrite taking into account that when the loadings are insignificant (for example <0.3) in a PC, there is not a clear relationships. There could be a close relationship in PC1, in PC2, in both or in none. The same applies for other figures.

Discussion and Conclusions

-          Lines 372-375. The correlationships between the preference variables seem to indicate that the panellists assessed all those variables as if they were an only variable, with redundant terms. This should have been amended using a proper test with separate samples for each variable.

-          Lines 378-379. Discuss here the differences in weight and the repercussion of the heating treatment on egg with different weights.

-          Line 387. The moisture content of the eggs should be included and discussed.

-          Lines 391-397. The text should be rewritten for clarity, or removed.

-          Rewrite to adjust to the experimental design and the results.

Author Response

(The authors gave the same response as above.)

Round 2

Reviewer 2 Report

A Comparison Between the Egg Yolk Flavor of Indigenous 2 Breeds and Commercial Laying Hens Based on Sensory Evaluation, Artificial Sensors and GC-MS

The document has been greatly improved, although there is still an issue with the term “replicate”:

Lines 140-142. Rewrite to make clear what a replicate is. Is it a farm or a cage?

Author Response

Thank you again for your scholarly comments on our manuscript. The comments provided are valuable and very helpful for revising and improving the quality of our paper. We have studied the comments carefully and have made the necessary corrections.
